# Toward a Symbolic AI Approach to the WHO/ACSM Physical Activity & Sedentary Behavior Guidelines

Carlo Allocca [1,*,†,‡], Samia Jilali [1,‡], Rohit Ail [1], Jaehun Lee [2], Byungho Kim [2], Alessio Antonini [3], Enrico Motta [3], Julia Schellong [4], Lisa Stieler [4], Muhammad Salman Haleem [5], Eleni Georga [6], Leandro Pecchia [5], Eugenio Gaeta [7] and Giuseppe Fico [7]

1.  Health Innovation, Samsung, Communications House, South St., Staines TW18 4QE, UK; s.jilali@samsung.com (S.J.); rohit.ail@samsung.com (R.A.)
2.  Samsung Research, 56 Seongchon-gil, Seoul 06765, Korea; jaehun20.lee@samsung.com (J.L.); bh1020.kim@samsung.com (B.K.)
3.  Knowledge Media Institute, The Open University, Milton Keynes MK7 6AA, UK; alessio.antonini@open.ac.uk (A.A.); enrico.motta@open.ac.uk (E.M.)
4.  Department of Psychotherapy and Psychosomatic Medicine, Faculty of Medicine, Technische Universität Dresden, 01307 Dresden, Germany; Julia.Schellong@uniklinikum-dresden.de (J.S.); Lisa.Stieler@uniklinikum-dresden.de (L.S.)
5.  School of Engineering, University of Warwick, Library Rd, Coventry CV4 7AL, UK; salman.haleem@warwick.ac.uk (M.S.H.); l.pecchia@warwick.ac.uk (L.P.)
6.  Department of Materials Science and Engineering, University of Ioannina, 45110 Ioannina, Greece; egeorga@cs.uoi.gr
7.  Life Supporting Technologies (LifeSTech), ETSI Telecomunicaciones, Universidad Politécnica de Madrid, Av. Complutense, 30, 28040 Madrid, Spain; eugenio.gaeta@upm.es (E.G.); gfico@lst.tfo.upm.es (G.F.)
*   Correspondence: c.allocca@samsung.com
†   Current address: Communications House, South St., Staines TW18 4QE, UK.
‡   These authors contributed equally to this work.



**Featured Application: Applied Sciences—Special Issue on Semantic Interoperability and Applications in Healthcare, 2021.**

**Abstract:** The World Health Organization and the American College of Sports Medicine have released guidelines on physical activity and sedentary behavior, as part of an effort to reduce inactivity worldwide. However, to date, there is no computational model that can facilitate the integration of these recommendations into health solutions (e.g., digital coaches). In this paper, we present an operational and machine-readable model that represents and is able to reason about these guidelines. To this end, we adopted a symbolic AI approach that combines two paradigms of research in knowledge representation and reasoning: ontology and rules. Thus, we first present HeLiFit, a domain ontology implemented in OWL, which models the main entities that characterize the definition of physical activity, as defined per guidance. Then, we describe HeLiFit-Rule, a set of rules implemented in the RDFox Rule language, which can be used to represent and reason with these recommendations in concrete real-world applications. Furthermore, to ensure a high level of syntactic/semantic interoperability across different systems, our framework is also compliant with the FHIR standard. Through motivating scenarios that highlight the need for such an implementation, we finally present an evaluation of our model that provides results that are both encouraging in terms of the value of our solution and also provide a basis for future work.

**Keywords:** symbolic AI; ontology; rules; WHO/ACSM physical activity guidelines; knowledge representation and reasoning

## 1. Introduction

Since the 1980s, the World Health Organization (WHO) and the American College of Sports Medicine (ACSM) have endorsed the role of physical activity to prevent and treat

noncommunicable diseases (NCDs), such as heart disease, stroke, diabetes, and cancers; to address risk factors such as hypertension, overweight, and obesity; and to improve mental health and overall quality of life and well-being [1]. As a matter of fact, WHO's Global Action Plan on Physical Activity seeks to reduce physical inactivity worldwide by 15 percentage points by 2030 [2]. To this end, WHO/ACSM have released new guidelines on physical activity and sedentary behavior concerning the amount and types of physical activity that offer significant health benefits and mitigate health risks [3].

Usually, these guidelines are provided as textual documents and each recommendation is expressed in natural language. As a result, it is not straightforward to integrate them into health solutions (e.g., digital coach). For example, let us consider the following example drawn from the guidelines [4]: *"When adults with chronic conditions or disabilities are not able to meet the key guidelines, they should engage in regular physical activity according to their abilities and should avoid inactivity"*. As we can notice, health professionals need to interpret the recommendation statement and adopt it to the individual's needs with his/her constraints, including physical activities preferences, contextual factors (e.g., environmental factors and personal factors), impairments from the body functions and structure, physical activity preferences, and any disorders or diseases [5]. In addition, compliance should be monitored, evaluated, and adapted to the individual's level of adherence and performance, on a regular basis. These are time-consuming activities, especially when aiming to monitor and support individuals at a worldwide scale. Consequently, these guidelines are often not applied, as they require significant time commitment by health professionals [6] and can also be too complex for elderly to follow without assistance [7–9].

We believe that a unified and coherent model for representing and reasoning about these recommendations can open the way to an effective integration of guideline management solutions with broader health applications (e.g., digital coach). To the best of our knowledge, to date, there is no computational model that can facilitate such an integration. Furthermore, the development of such a model is a challenging task, as it needs to deal with a number of issues, including: (i) establishing a shared and validated interpretation of the guidelines, considering the health professional perspective; (ii) quantifying, measuring, and combining different types of physical activities with different intensity, duration, etc; (iii) checking adherence to the recommendations over a given period of time; and (iv) re-evaluating and adapting recommendations to the user, whether or not there is adherence.

The purpose of this paper is to address the above practical barriers and challenges by providing a reusable, operational, and machine-readable model that allows not only to represent computationally the guidelines but also to reason about them. To this end, we propose a symbolic AI approach that combines two main paradigms of knowledge representation and reasoning (KR&R): ontology and rules [10]. With respect to the first, we present HeLiFit, a domain ontology implemented in the Ontology Web Language (OWL), which models the key concepts and properties that characterize the notion of physical activity, as required by the WHO and ACSM. With respect to the second, we describe HeLiFit-Rule, a set of rules based on HeLiFit, implemented in the RDFox Rule language, to be used to express and reason about these guidelines in real-world health applications. In addition, to ensure a high level of syntactic and semantic interoperability, when integrating our solution with different systems, our framework is also compliant with the Fast Healthcare Interoperability Resources (FHIR) data model. As a result, such a model is configured as a solution that can be adopted as a plug-and-play module, when developing health-related systems, e.g., digital coach, which are intended to monitor users and and provide health recommendations to them [11]. Furthermore, another advantage of the proposed approach is that, being based on HeLiFit and HeLiFit-Rule, where one or more rules encode a particular element of the guidelines, is able to accommodate changes or extensions to the recommendations from WHO and/or ACSM in a modular way, with no need for an overall redesign or retraining of the model.

In a nutshell, the key contributions of this work are the followings: (a) we identify real-world reference use cases and application scenarios motivating the needs of the proposed

work; (b) we introduce HeLiFit, a domain ontology implemented in OWL that provides a model, in term of concepts and properties, of the domain of interest; (c) we describe HeLiFit-Rule, a set of rules representing recommendations for physical activity, as per WHO/ACSM guidelines; (d) we describe an evaluation with domain experts, in terms of our system's ability to model user compliance and performance and adapt its advice accordingly.

The rest of this paper is organized as follows: in the Materials and Methods section, we discuss the reference use cases and application scenarios motivating the needs addressed by the proposed model. We examine the main gaps with respect to the existing literature, when it comes to representing these guidelines, and we present our approach in detail. In the Results section, we describe the implementation of HeLiFit in OWL, and HeLiFit-Rule in the RDFox Rule language. Furthermore, we describe the evaluation of our model, which was performed with domain experts, and we illustrate how our model was used to create a knowledge graph encoding specific user data. In the Discussion section, we elaborate upon the main achievements, the limitations of our approach and the main directions for future work. Finally, in the Conclusion section, we recap and summarize our work.

## 2. Materials and Methods

In this section, we first describe the reference use cases and application scenarios as elaborated in the EU GATEKEEPER project (GATEKEEPER is a European Multi-Centric Large-Scale Pilots on Smart Living Environments with one of the main objectives to deliver AI-based services for early detection and prevention of chronic diseases.) (Section 2.1). Then, we examine the main gaps with the existing literature when it comes to the implementation of these guidelines, especially from a computational point of view (Section 2.2). Finally, we present our symbolic AI approach, based on ontology and rules (Section 2.3).

### 2.1. Reference Use Cases and Motivating Application Scenarios

2.1.1. Lifestyle-Related Early Detection and Interventions

We all are living increasingly longer [12]. In Europe, life expectancy at birth for males will be 84.6 years by 2060–2065, compared to 76.6 in 2015, whereas for females, it will be 89.1 years by 2060–2065, compared to 82.5 in 2015 [12]. Thus, the focus of these reference use cases is based on two main unavoidable aspects that most elderly people will eventually experience: frailty and sedentary behavior and/or issues related to mental health and well-being.

*Frailty* is considered to be a common clinical syndrome in older adults that carries an increased risk for poor health outcomes, including falls, incident disability, and hospitalization [13]. It is usually associated with low levels of physical activity [14], which especially increases the mortality risk in older people [15].

The issue of mental health and well-being is considered in the context of older adults living with chronic conditions that have unmet care needs related to their physical and psychological health, social life, stresses of life, as well as the environment in which they live. It is also well known that elderly people wish to live independently for as long as possible [5]. Therefore, a biopsychosocial and person-centred approach to health care is needed [16].

The proposed semantic framework, which is composed by HeLiFit and HeLiFit-Rule, can be integrated into concrete real-world health applications (e.g., digital coach), in order to address health care needs related to improving physical activity and, consequently, mental health. As a result, it can reduce the effort required from health care professionals.

2.1.2. Health Reasoning and Explainable AI (XAI): Establishing a Shared and Validated Interpretation

Let us consider the following two examples, where, in contrast with Sara, Juan is not compliant with the WHO guidelines:

- **[Not compliant:]** Juan, 25 years old, wearing a smart watch, has performed this week the following activities: On Wednesday, he did yoga from 8 a.m. to 8:45 a.m., which

is equivalent to 2025 steps and on Friday, he went for a run from 7 a.m. to 8 a.m., in total, 6000 steps were recorded by the watch.

- **[Compliant:]** Sara, 35 years old, wearing a Google smart watch, this week performed the following activities: On Monday, she played basketball from 6 p.m. to 7:30 p.m. and 13,500 steps were recorded by the watch; on Thursday afternoon, she went running from 8 a.m. to 9:30 a.m. and she did 25,000 steps, as counted by the watch. On Friday, she played handball from 2 p.m. to 3 p.m., with the watch shoowing 20,880 steps recorded. On Saturday and Sunday, she did weight lifting from 2 p.m. to 2:30 p.m.

In this respect, the key questions for our computational solutions are Q1: How to assess and integrate the different physical activities that Sara and Juan performed over a week period? Q2: How to check automatically whether their levels of activity are adhering to the WHO/ACSM recommendations? Q3: In both cases (yes or no), what are the additional recommendations that they can take advantage of to improve their health? The approach that is presented in this paper can be exploited for answering these questions.

Currently, a domain expert is required to analyze each data source separately to derive conclusions on physical activity regime and recommendation [17]. The framework presented in this paper can support the automatization of the underlying process from various perspectives. First, it can be used to integrate the information coming from different data sources and build a coherent knowledge graph (through a materialized or virtual approach [18]). Second, it can be used to support the above medical reasoning as it is capable to represent, combine, and reason upon the performed activities. Finally, as a symbolic-AI-based approach, it provides the means for explaining the rationale for the decisions taken in relation to the provided recommendations [19].

### 2.1.3. Digital Coaching in the Health Care Domain

Digital coaching, as a field of research in the health care domain, has the primary goal of improving personalized patient engagement and adherence, both of which are necessary for achieving long-term behavioral changes and adoption of a healthier lifestyle [20]. According to [21,22], the success of this type of systems is based on their capacity of adaptation, which is the notion of tailoring a communication on the basis of external information [22] and involves attempts to increase attention or motivation by conveying personalized communication [22].

Our approach can be utilized to better support the above aspects by recognizing the degree of adherence to the WHO/ACSM recommendations and by improving the identification of appropriate recommendations on which to implement mechanisms for adaptation, user targeting, and context awareness.

### 2.2. Computer-Interpretable Guidelines

In 2019, WHO initiated a revision of its 2010 guidelines on physical activities and sedentary behavior. The revision aimed to be aligned with the guidelines with the latest evidences and clinical studies on the field. They also provided expert recommendations on the optimal amount of physical activity in children, adolescents, adults, and elderly people, (>64 years), as well as pregnant and postpartum women, and people living with chronic conditions or disabilities [23]. Alongside the revision of the WHO guidelines, there is a growing interest in incorporating WHO and other similar guidelines into information technologies. For instance, the Physical Activity Ontology (PACO) [24] focuses on the interoperability of physical activity data, while the HeLiS [25] ontology combines physical activity and nutrition for monitoring unhealthy behaviors. However, so far, no other works actually materialize the new WHO guidelines into a service. In general, the use of clinical guidelines in health care information system and health-related activities concerns a wide range of applications. For example, "Computer Interpretable Clinical Guidelines" [26] (CICG) is largely focused on interoperability and management of guidelines (i.e., hierarchies and adaptation to local clinical protocols), execution engines, and decision support systems for clinicians. Here, it is worth mentioning the Guideline Interchange Format (GLIF) in

its third iteration, GLIF3 [27], and the GLIF Execution Engines, GLEE [28]. The GLIF ecosystem is tailored to the life cycle and consumption of clinical guidelines by health care institutions, from design, encoding, and validation to local adaptation, integration, application, and revision. This combination is used to automatise data and clinical actions, using the guidelines to define clinical processes [29] and the interpretation of health care records [30]. This direction is promising and, overall, in support of a systematic production and revision of reusable and validated guidelines.

A central application of this field are decision support systems, e.g., for coordinating health care services (personnel and facilities) [31] and for mitigating the risks of medical errors [32]. Moreover, CICG and logical reasoning have been applied to the construction and maintenance of corpora of guidelines, e.g., to identify and lower the risk of interference between different guidelines [33,34], and to support authoring and formal verification of guidelines [35].

To sum up, systems for CICG are growing from both a technical and application point of view. This trend connects the production of machine-readable guidelines with the clinical processes of adopting, adapting, and revising guidelines, outlining a sustainable and flexible approach, firmly grounded on clinical standards. However, the development of these systems is still oriented to a strictly clinical domain, leaving outside their scope well-being and consumer applications, such as digital coaching.

### 2.3. Symbolic AI Approach: From Analysis to Design

In this section, we present a symbolic AI approach that combines two main paradigms of the knowledge representation and reasoning field, i.e., ontology and rules, to implement a reusable, operational, and machine-readable model that represents and is able to reason about the WHO/ACSM guidelines. At the most general level, our approach is based on the elements as illustrated in Figure 1:

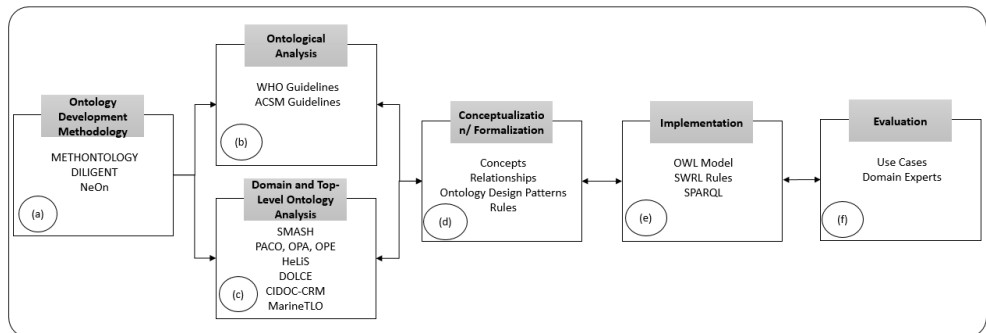

**Figure 1.** The approach step by step.

(a)  An analysis of ontology development approaches, including METHONTOLOGY [36], DILIGENT [37], NeOn [38] and the one proposed by Gangemi et al. in [39] and implemented by Allocca et al. in [40].

(b)  An ontological analysis of the main concepts and relationships that are involved in the domain of Physical Activity & Sedentary Behavior Guidelines [3].

(c)  An analysis of existing domain ontologies, including PACO [24], HELIOS [25], and OPA [41], to reuse as much as possible existing concepts and relationships. At the same time, we also analyzed top-level ontologies, such as DOLCE [42], CIDOC-CRM [43], and SUMO [44], to maximize interoperability and facilitate a design approach based on ontology patterns.

(d)  The design of an appropriate domain ontology—concepts and properties—to model what WHO/ACSM requires in relation of physical activity and the formulation of the recommendations as IF-THEN rules.

(e)  The implementation of an ontology (HeLiFit) using OWL—Ontology Web Language—and of the set of rules (HeLiFit-Rule) using RDFox Rule Language.

(f) A validation process centred on checking the logical consistency of the ontology and evaluating it in terms of appropriateness and usefulness, as determined by domain experts.

### 2.3.1. Ontology Development Methodology

As shown in Figure 1, in the step labeled as (a), our approach started with an ontology development process. This provides a life cycle design and development methodology, split in well-defined steps, that can be continuously applied to model the domain of discourse in a systematic manner [40]. To this end, we compared the most used methodologies in the literature. In particular, we contrasted METHONTOLOGY [36], DILIGENT [37], and Neon [38], and selected METHONTOLOGY. In contrast with the other approaches, METHONTOLOGY puts emphasis on a centralized engineering process, which is the one relevant to our scenario, as the knowledge was acquired from WHO guidelines and validated directly with the domain experts from the GATEKEEPER EU project. Thus, we continue the description of our approach by elaborating the various steps of the methodology, which include ontological analysis, domain and top-level ontology analysis, conceptualization and formalization, implementation, and evaluation.

### 2.3.2. Ontological Analysis

We aim at understanding what we need to represent in terms of the entities of our domain. To this end, as reported in Figure 1 (the step labeled as (b)), we performed an ontological analysis of the Physical Activity & Sedentary Behaviour Guidelines (WHO and ACSM). To exemplify, we show in Figure 2 a typical example of WHO/ACSM recommendation. In general, guidelines associate a recommendation to a target group of people. In this view, we identify two main components: target audience and recommendation. The target audience defines a group of people in terms of a set of shared characteristics—in the example, having an age between 18 and 64 years old. These definitions refer to properties that change over time, such as age or conditions (e.g., pregnancy or postpregnancy). In other words, these guidelines define transient rather than static groups (e.g., by ethnicity or gender).

**Adults:** For all adults in the range of 18-64 years,

1. At least 150-300 minutes per week of moderate-intensity aerobic physical activity; or
2. At least 75-150 minutes per week of vigorous-intensity aerobic physical activity; or
3. An equivalent combination of moderate and vigorous-intensity activity throughout the week for substantial health benefits.

**Figure 2.** Example of WHO recommendation [3].

The recommendation describes a physical activity as a set of characteristics (see Figure 3) involving modality, intensity, frequency, and duration. The WHO guidelines do not provide a list but specify modality, in the example, as an "aerobic" activity, as a class of exercises. In this regard, the use of the guidelines requires filling a gap in terms of mapping the specific activity observed to the relevant modality—e.g., fast-walking to "aerobic", yoga to "muscle-strengtening" activities. The guidelines also specify a level of intensity (e.g., "moderate" or "vigorous" intensity), which provides a qualitative evaluation of the effort spent by a person on the activity. Specifically, WHO defines intensity in terms of multipliers of energy consumption at rest—e.g., low-intensity is when an activity consumption is less than three times resting.

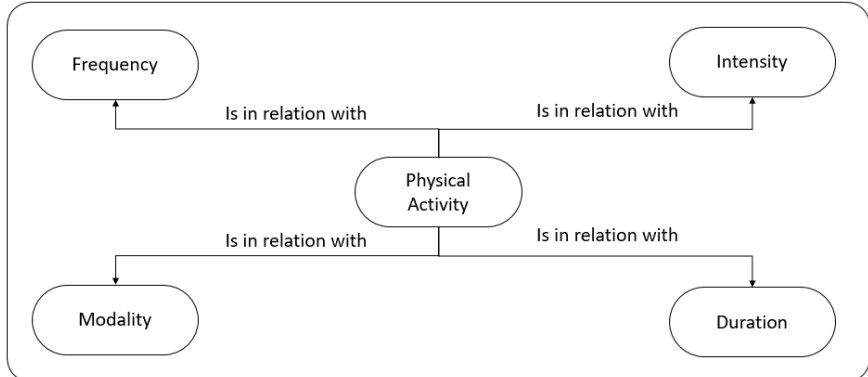

**Figure 3.** Physical activity components.

Concerning frequency, WHO guidelines refer to a minimum number of sessions per week (e.g., three times or once a week). In this view, the duration (e.g., 150–300 min) is cumulative of all sessions in a week and expressed in the guidelines as a requirement for a minimum number of minutes per week or a range (with a maximum). It is worth noticing that the WHO definition of duration is not a single value plus/minus a variation, but a range of equally valid durations, depending on the intensity.

It is also worth pointing out that the WHO guidelines do not specify ontological and contingent constraints related to the combination of modality, intensity, frequency, and duration. For instance, an aerobic activity has a minimum duration and therefore, implies an upper bound to the frequency, in terms of maximum number of aerobic activities of minimum duration, which can be carried out in the specified time period. Similarly, the frequency, as minimum number of sessions per week, implies a lower bound to their duration. Duration can be unbounded but, relying on external knowledge, the implementation of the guidelines may include an explicit upper bound. Different from fitness ontologies, the WHO does not qualify the settings of the exercise such as the elevation, peace, or temperature of the exercise location. Indeed, a fitness ontology may focus on exercising a specific muscle, whereas the WHO guidelines are concerned with the overall effect on the person. In this view, WHO guidelines provide a constraint-centred description of an optimal weekly regime. Another level of flexibility concerns the possibility to combine different physical activities. However, WHO does not specify a metric to evaluate combinations. Hence, the realization of the guidelines into a scheduler system also requires the inclusion of additional sources of domain knowledge. Finally, getting inspired also by the Neon methodology [38], in Table 1, we describe the HeLiFit specification in terms of its purpose, scope, implementation language, target users, and intended use.

In order to provide a set of definitions as reference, we report here those for physical activity, physical inactivity, sedentary behavior, and exercise in accordance with [45,46] that the HeLiFit ontology should support. In particular, we describe them as follows:

1. Physical activity—a range of waking behaviors that share the common feature of increasing energy expenditure that is determined, for a given activity, by the intensity, duration, and frequency of muscular movement.
2. Physical inactivity—the failure to achieve the minimum activity recommendations for health.
3. Sedentary behavior—sedentary behavior refers to any waking behavior characterized by an energy expenditure less or equal to 1.5 METs while in a sitting, reclining, or lying posture.
4. Exercise—a form of physical activity that is planned, structured, and repetitive with the aim of improving or maintaining fitness.

With the above, we have set the scope of our ontology and the ontological analysis of the domain of discourse. In the following, we proceed with figuring out how to model it, starting from what already exists.

**Table 1.** HeLiFit ontology and rules specifications.

| | |
|---|---|
| Intensity category | The purpose of HeLiFit and HeLiFit-Rule is to represent and reason upon the WHO/ACSM guidelines on physical activity and sedentary behavior. |
| Scope | The HeLiFit ontology should focus on characterizing the notion of physical activity as used in the context of WHO/ACSM guidelines and allowing a mean to measure it, whereas HeLiFit focuses on reasoning over the different levels of adherence and issue recommendations to users. |
| Implementation language | Web Ontology Language (OWL) for *HeLiFit* and RDFox Rule Language for HeLiFit-Rule. |
| Target user | The primary target users are health care professionals (User 1) working with aging frailty users (user 2), aiming to recommend or coach them on physical activity and exercise. Another group of target users (user 3) is professionals involved in the development of digital coach solutions to support physical activity and exercise recommendation, i.e., <br> User 1—Health care professionals dealing with frailty <br> User 2—Aging frailty users trainer <br> User 3—Software developer or researcher working in the domain of digital coach for adherence in physical activity domain |
| Intended uses | User 1: The intended uses of the ontology include: (1) supporting the process of physical activity regimen recommendation to a patient and (2) modifying current physical activity regimen for better adherence. <br> User 2: The intended use is to support the process of modification of user adherence based on changes in the user profile. <br> User 3: The intended use is to support the development of personalized solutions for physical activity maintenance and modification based on the user's need. |

### 2.3.3. Domain and Top-Level Ontology Analysis

We aim at understanding what already exists in the literature, which can help us to model the entities of our domain and, at the same time, to identify the gaps, Figure 1, see step labeled as (c)). To this end, we compared the most relevant domain ontologies, including SMASH (Semantic Mining of Activity, Social, and Health data) (https://bioportal.bioontology.org/ontologies/SMASHPHYSICAL, accessed on 28 November 2021), OPA [41], OPE [47], PACO [24], HeLiS [25], as well as top-level ontologies, including DOLCE [42], CIDOC-CRM [43], and SUMO [44], to ensure a systematic, pattern-based development of our model.

**Domain Ontologies:** SMASH (https://bioportal.bioontology.org/ontologies/SMASHPHYSICAL, accessed on 28 November 2021), which focuses on describing the semantic features of health care data and social networks, provides a well-developed physical activity type hierarchy that is divided into athletic sports, exercise, and occupational activity. The main drawback is the limited (or missing) coverage of relevant concepts, such as intensity and amount of physical activity, which are required for our case. OPA [41], the ontology for assessing physical activity and sedentary behavior, provides a baseline for characterizing physical activity, sedentary behavior, and the context in which it occurs, including factors such as space, time, weather, and social. Although it provides some

relevant concepts to model physical activity, including *TemporalEntity*, *SpaceEntity*, *IntensityOfActivity* and *Anthropometry*, the concepts needed for formalizing WHO rules are only partially covered. These include notions such as VO2MAX, HRR, MET, and others, which are captured through personal sensor devices (wearables and/or smart phone) and are of primary importance when dealing with the high-resolution temporal data associated with physical activity. OPE (https://bioportal.bioontology.org/ontologies/OPE, accessed on 28 November 2021) (the Ontology of Physical Exercises) [47] provides a reference for describing an exercise in terms of functional movements, emphasizing the involvement of the *Musculoskeletal* and *Muscle* parts when performing a specific type of fitness: exergaming. Therefore, OPE has quite a few limitations in representing nongame-based physical activities with sufficient detail. PACO [24] (Physical Activity Ontology) supports the structuring and standardizing of heterogeneous descriptions of physical activities to address semantic interoperability. It has been built by extracting concepts related to physical activity from medical corpora, including questionnaires and assessment scales. PACO, with 225 classes, 20 object properties, 1 data property, and 23 instances, includes the notion of exercise leisure activity and a number of modifiers, such as *Amount*, *Frequency*, and *Intensity*. However, it does not provide a classification of the activities in aerobic and anaerobic ones, and does not provide classes and proprieties to measure and estimate the intensity of an activity (e.g., VO2MAX, HRR, MET), which are needed for formalizing WHO rules. As far as HeLiS (Healthy Lifestyle Support ) is concerned [25], it aims at modelling foods and nutrients as well as physical activities. However, its primary focus is on the specific foods individuals must consume, rather than the contextual physical activity elements that are needed to personalize the WHO recommendations. Nevertheless, all the above ontologies have inspired technical aspects of the engineering of the HeLiFit ontology, as per relevant overlapping objectives, e.g., the structure of physical activities and performance context.

**Top-Level Ontologies:** (Refer to [48] for a full survey of Top-Level Ontologies) With the aim of connecting our work to a top-level ontology, we have examined the most relevant three: DOLCE [42], CIDOC-CRM [43], and SUMO [44]. The first one, DOLCE, is oriented toward capturing the ontological categories underlying natural language and human common sense, providing high level classes, such as *Event*. This encompasses at least one *Agent* that *isParticipantIn* it, and executes a *Task*, which typically *isDefinedIn* a *Plan*, *Workflow*, or *Project*. The second one, CIDOC-CRM, focuses on representing event based cultural heritage and contains generic upper classes, such as *Space-Time*, *Events*, *Activity*, and *Measurement*. Finally, the third one, SUMO (Suggested Upper Merged Ontology), is a formal ontology and it is defined in the higher order logical language of SUO-KIF. It includes dozens of domains ontologies, contains roughly 20,000 terms and 80,000 logical statements and it is largely used for translations to languages and mappings to WordNet [44]. We suggest [49] for a deeper overview of existing foundational ontologies and how they are used across several computer-based tasks.

In conclusion, although all these three ontologies could be in principle adopted as our top level ontology, we selected CIDOC-CRM, as it provides an ontology design pattern that can be easily extended for use in the context of modeling WHO/ACSM recommendations.

### 2.3.4. Conceptualization and Formalization

We aim at conceptualizing what is not covered by existing works and is required to be able to combine different physical activities, to check adherence and to provide recommendations according to the WHO/ACSM guidelines. To this end, as reported in Figure 1 (the step labeled as (d)), we capitalized on the outcome of the ontological analysis of the guidelines (both WHO and ACSM) and we proceeded with the engineering of our ontology, HeLiFit, and set of rules, HeLiFit-Rule.

*HeLiFit* **ontology.** As shown in Figure 4, we identified the general top level categories and the *Activity-Temporal-Entity-Pattern* from CIDOC-CRM to capture the semantics of the physical activity as an event which happen over a limited extent in time and to model all

the other measurements associated with it, including intensity, frequency, modality, and duration (see Figure 3 for more details).

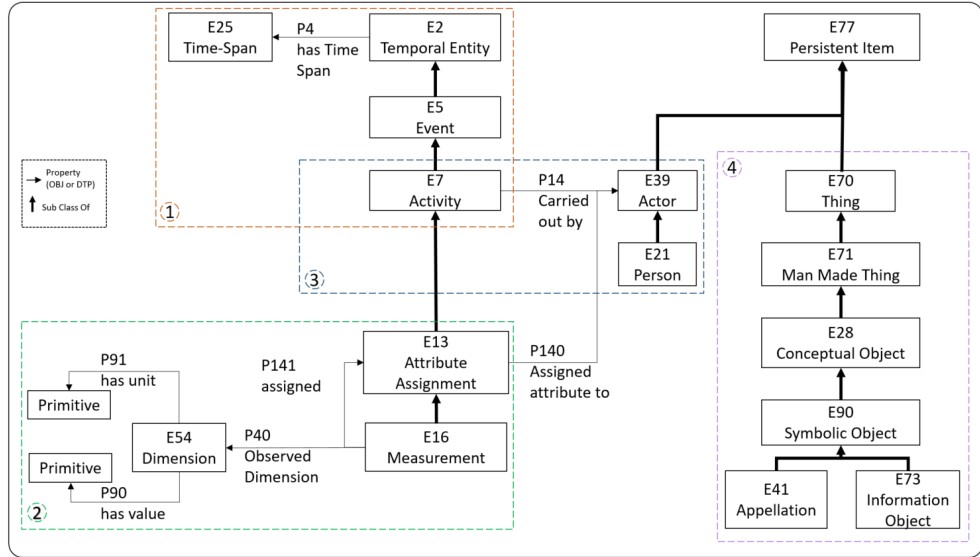

**Figure 4.** Activity-Temporal-Entity-Pattern from CIDOC-CRM.

Figure 4 illustrates four main blocks. Block 1 is used to capture the physical activity that is considered as a temporal entity with start and end time. Block 2 is to model all quantifiable properties that can be measured and that have a value and unit such as duration and height. Block 3 illustrates the relation between the activity and the person; in fact, a physical activity is carried out by a person. Block 4 is used to model the recommendations, where each recommendation is identified by a unique code under E41 Appellation. Based on the *Activity-Temporal-Entity-Pattern*, we have proceeded with its extension to model all the domain specific concepts and properties including physical activity, exercise, cardio-aerobic, etc. and all the relevant measurements when performing and recording it, as required by the WHO and ACSCM guidelines. The top-level classes in our model reflect the concepts and properties as reported in Figure 4 (see Section 2.3.4) from CIDOC-CRM [43] ontologies. Specifically, top-level classes consist of the classes such as *Activity-Temporal-Pattern* that comprises all sets of phenomena, such as periods, events and states, ecosystems, which are bounded in time and space, and observable entity that comprises the behavior and interaction of physical things being observed through events or activities or state, either directly by human sensory impression, or enhanced with tools and measurement devices. Based on it, we illustrated in Figures 5 and 6 how the main the CIDOC-CRM-based patterns are extended to capture the underlying domain knowledge. In particular, Figure 5 shows the extension of the activity (E7 Activity) to model the *Actor-Activity-Performance-Pattern* that schematizes when a person performs one or more physical activity and, for each of them obtains a set of performance parameters—e.g., duration, step counts, VO2Max, and so on—that are modeled as subclass of *E54 Dimension* that comprises quantifiable properties that can be measured by some calibrated means and can be approximated by values. Likewise, in Figure 6 shows the extension of the the measurements (E16 Measurement) to be able to model the *Actor-Anthropometry-Vital-Signs-Measurements-Pattern* that schematizes when a person performs either an Anthropometric or a vital signs measurements.

As shown in Figure 4, we identified the general top level categories from CIDOC-CRM to capture the semantics of the physical activity and model the relevant measurements, including intensity, frequency, modality, and duration (see Figure 3 for more details). These top-level classes reflect the concepts and properties of the CIDOC-CRM ontology. Specifically, they consist of classes that allow us to: (1) model the physical activity as a temporal entity which is bounded in time and space; (2) describe the patient as a persistent item who has the potential to perform intentional actions for which they can be held

responsible, such as performing a physical activity; the patient is linked to the physical activity by means of the object property carried out by; (3) collect all the measurements characterized by dimensions as quantifiable properties of the physical activity (duration, frequency, . . . ) and the patient (age, height, . . . ) that can be approximated by values and have units; (4) model the recommendations that the patient is subject to, as a symbolic object where each recommendation has a unique *Appellation* (WHOcode001, WHOcode002 . . . ).

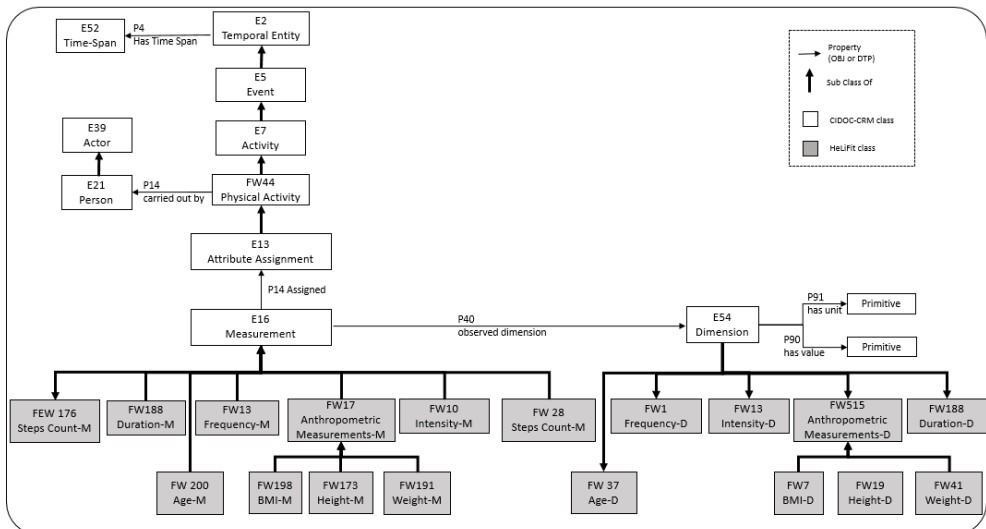

**Figure 5.** Part of classes of HeliFit-*Actor-Activity-Performance-Pattern.*

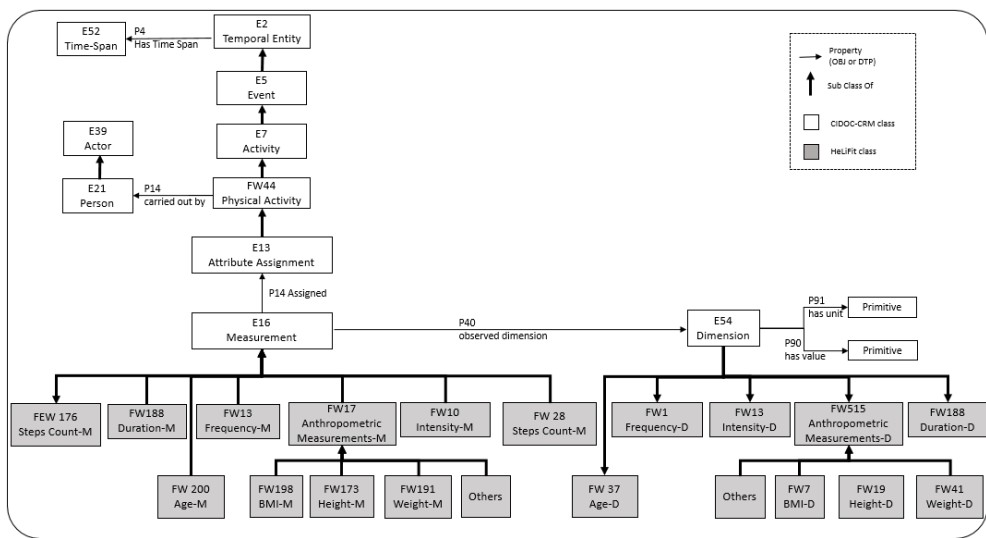

**Figure 6.** Part of classes of HeliFit-Actor-Anthropometric-Vital-Signs-Measurements.

Based on these patterns, we have proceeded with their extension to model all the domain specific concepts and properties, including physical activity, exercise, cardio-aerobic, etc. and all the relevant measurements when performing and recording it, as required by the WHO and ACSCM guidelines. In Figures 5 and 6, we illustrate how the main the CIDOC-CRM-based patterns are extended to capture the underlying domain knowledge. In particular, Figure 5 shows the extension of the activity (E7 Activity) to model the *Actor-Activity-Performance-Pattern* that schematizes when a person performs one or more physical activities and, for each of them, obtains a set of performance parameters—e.g., duration, step counts, VO2Max, and so on. These are modeled as subclasses of *E54 Dimension*, which comprises quantifiable properties that can be measured by some calibrated means and approximated by numerical values. Likewise, Figure 6 shows the

extension of the the measurements (E16 Measurement) to be able to model the *Actor-Anthropometry-Vital-Signs-Measurements-Pattern* that schematizes when a person performs either an Anthropometric or a vital signs measurements.

It is relevant to highlight here that HeLiFit is not supposed to be a single ontology covering the entirety of what exists w.r.t. physical activity. Instead, it aims to formalize a domain model that provides suitable abstractions of the domains under consideration, to cover the use cases of fraitly, mental health, and physical activity, and to enable computational support for issuing recommendations, in accordance with the guidelines published by WHO and ACSM.

*HeLiFit-Rules.* HeliFit-Rules are the set of rules used for formalizing and triggering recommendations. These were implemented using the RDFox Rule Language—a declarative logic-based language of type IF-THEN, which is part of the RDFox engine for high-performance knowledge graph and semantic reasoning [50].

In this section, we will describe the HeliFit-Rules in detail by first introducing a set of rules from the WHO/ACSM guidelines and then presenting the relevant rule representation in terms of the RDFox Rule Language.

Table 2 presents a subset of WHO recommendations for the four categories, namely children and adolescents, adults (aged 18–64 years), older adults (aged 65 years and older), and pregnant and postpartum women.

Next, we discuss HeLiFit-Rule, the implementation of WHO and ACSM recommendations, which relies on the HeLiFit ontology and is based on the Rule paradigm [10]. One of the first challenges was to address the comparison of the performances of different physical activities and issue a strategy to specify a specific physical activity that all the others can be reduced to. In other words, we want to be able to state that performing physical activity X1 for a duration Y1 and intensity Z1 is equivalent to performing physical activity X2 for a duration Y2 and intensity Z2. We do this by reducing all physical activities to walking and to a number of steps [51]:

1. **Rule to convert a physical activity into steps:** In order to make physical activities comparable, and hence apply aggregation operations when more that one physical activity is performed over a specific time window, we convert them into steps using the duration and an activity-specific conversion factor, as shown in Equation (1).

$$\text{Number of steps} = D \times \alpha \qquad (1)$$

   $D$ specifies the duration in minutes of the physical activity performed while $\alpha$ is a coefficient that depends on the physical activity. For example, 30 min of baseball, which has a coefficient of 150, is equivalent to $30 \times 150 = 4500$ steps. This conversion is formulated in Figure 7.

**Figure 7.** Example of rule to convert a physical activity into steps.

**Table 2.** WHO Recommendations set.

| **Children and Adolescents** |
|:---:|
| Children and adolescents should do at least an average of 60 min per day of moderate- to vigorous-intensity, mostly aerobic physical activity across the week. |
| Vigorous-intensity aerobic activities, as well as those that strengthen muscle and bone, should be incorporated at least 3 days a week. |
| **Adults (aged 18–64 years)** |
| All adults should undertake regular physical activity. |
| Adults should do at least 150–300 min of moderate-intensity aerobic physical activity; or at least 75–150 min of vigorous-intensity aerobic physical activity; or an equivalent combination of moderate- and vigorous-intensity activity throughout the week, for substantial health benefits. |
| Adults should also do muscle-strengthening activities at moderate or greater intensity that involve all major muscle groups on 2 or more days a week, as these provide additional health benefits. |
| Adults should limit the amount of time spent being sedentary. Replacing sedentary time with physical activity of any intensity (including light intensity) provides health benefits. |
| **Older adults (aged 65 years and older)** |
| Older adults should do at least 150—300 min of moderate-intensity aerobic physical activity; or at least 75–150 min of vigorous-intensity aerobic physical activity; or an equivalent combination of moderate- and vigorous intensity activity throughout the week, for substantial health benefits. |
| Older adults may increase moderate intensity aerobic physical activity to more than 300 min; or do more than 150 min of vigorous-intensity aerobic physical activity; or an equivalent combination of moderate- and vigorous intensity activity throughout the week, for additional health benefits. |
| To help reduce the detrimental effects of high levels of sedentary behavior on health, older adults should aim to do more than the recommended levels of moderate to vigorous-intensity physical activity. |
| **Pregnant and postpartum women** |
| All pregnant and postpartum women without contraindication should do at least 150 min of moderate-intensity aerobic physical activity throughout the week for substantial health benefits. |
| All pregnant and postpartum women without contraindication should incorporate a variety of aerobic and muscle-strengthening activities. Adding gentle stretching may also be beneficial |
| Pregnant and postpartum women should limit the amount of time spent being sedentary. Replacing sedentary time with physical activity of any intensity (including light intensity) provides health benefits. |

2. **Rule to compute the intensity of a physical activity:** The intensity of a physical activity is related to how hard our body works while doing a specific physical activity [51]. There are four levels of the physical activity intensity: *High*, *Vigorous*, *Moderate*, and *Sedentary* and they are measured and estimated by one of the variables: Heart rate reserve ($HRR$), rate reserve max ($HR_{max}$), metabolic equivalents ($MET$), maximal oxygen consumption ($VO_{2max}$), and steps count. In Table 3, we present the different rules to estimate the physical activity intensity.

**Table 3.** Level of Intensities of Physical Activity.

| Intensity Category | Objective Measures |
|---|---|
| Sedentary | $MET < 1.6$<br>$HR_{max} < 40\%$<br>$HRR < 20\%$<br>$VO_{2max} < 20\%$<br>$Steps < 119$ per minute [51] |
| Moderate | $3 < MET < 6$<br>$55\% < HR_{max} < 70\%$<br>$40\% < HRR < 60\%$<br>$40\% < VO_{2max} < 60\%$<br>$119 < Steps < 123$ per minute [51] |
| Vigorous | $6 < MET < 9$<br>$70\% < HR_{max} < 90\%$<br>$60\% < HRR < 80\%$<br>$60\% < VO_{2max} < 80\%$<br>$137.8 < Steps < 140.7$ per minute [51] |
| High | $9 < MET$<br>$90\% < HR_{max}$<br>$80\% < HRR$<br>$80\% < VO_{2max}$<br>$Steps > 140.7$ per minute [51] |

The WHO and ACSM guidelines are structured for four main categories of users: (i) children and adolescents (aged 5–17 yeas old); (ii) adults (aged 18–64); (iii) older adults (aged 64 years and older), and (iv) pregnant women. HeLiFit-Rule implements 137 rules to cover all the above users. In the paper, we describe a few of them and leave the rest in the Appendix A. Figure 8 below models the case of a physical activity with low intensity that is classified as Sedentary.

**Figure 8.** Rule to classify the user as sedentary.

3. **Rule to issue WHO/ACSM recommendations:** Rules are triggered according to the user's age, the frequency, the modality, the duration (typically a week), and the intensity of the physical activity. While frequency, modality, and duration are given explicitly, intensity needs to be calculated. Once we have computed the intensity of all the physical activities that a user has performed over a specific time window, using the above rule, we aggregate them and, based on the results, one or more recommendations are triggered. As an example, we show the rule that codifies the following WHO guideline:

   **WHO Recommendation**: *Adults(18–64 years) should also do muscle-strengthening activities at moderate or greater intensity that involve all major muscle groups on 2 or more days a week, as these provide additional health benefits.*

   The user, assumed to be between 18 and 65 years, will get this recommendation in two cases:

**Case 1:** she/he performed any muscle-strengthening physical activity but the intensity of the physical activity is lower than moderate or the frequency is less than two. This case is covered in Figure 9 below:

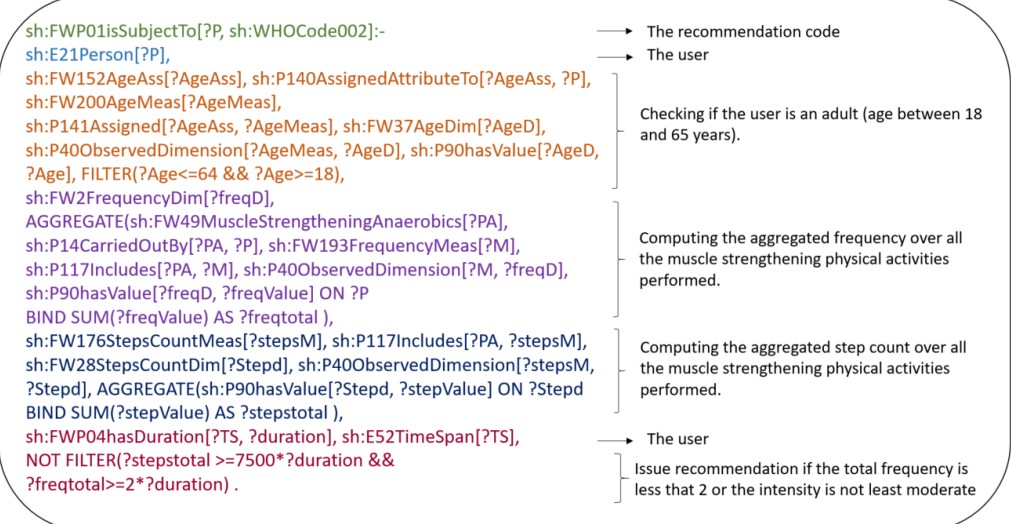

**Figure 9.** Rule to provide recommendation based on the frequency, modality, and intensity of the physical activities performed.

**Case 2:** she/he did not do any muscle-strengthening physical activity. This case is covered in Figure 10:

```
sh:FWP01isSubjectTo[?P, sh:WHOCode002]:-
sh:E21Person[?P],
sh:FW44PhysicalActivity[?PA], sh:P14CarriedOutBy[?PA, ?P],
sh:FW152AgeAss[?AgeAss], sh:P140AssignedAttributeTo[?AgeAss,?P],
sh:FW200AgeMeas[?AgeMeas], sh:P141Assigned[?AgeAss, ?AgeMeas],
sh:FW37AgeDim[?AgeD], sh:P40ObservedDimension[?AgeMeas, ?AgeD],
sh:P90hasValue[?AgeD, ?Age], FILTER(?Age<=64 && ?Age>=18),
NOT EXISTS ?PA IN sh:FW49MuscleStrengtheningAnaerobics[?PA] .
```

- → The recommendation code
- → The user
- → The user performed physical activity
- Checking if the user is an adult (age between 18 and 65 years).
- → Checking if the user has performed muscle strengthening activities

**Figure 10.** Rule to check if the user has performed any muscle-strengthening activities.

Figures 9 and 10 cover the cases in which a user does not adhere to the guidance. However, HeLiFit-Rule implements also the case when the user does, getting a compliance feedback: "Well done, You are compliant with WHO rule that Adults should also do muscle-strengthening activities at moderate or greater intensity that involve all major muscle groups on 2 or more days a week". These positive feedbacks provide additional health benefits.

As an example of positive feedback, we show in Figure 11 below:

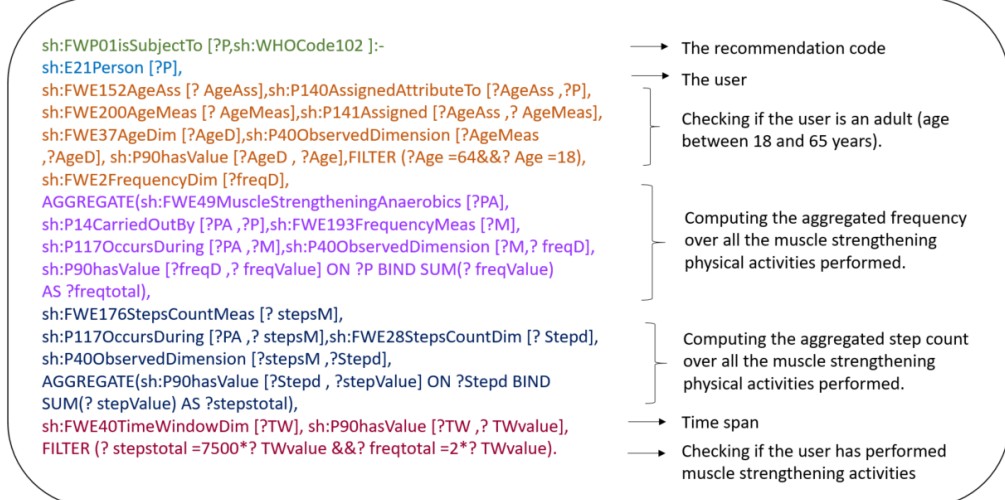

**Figure 11.** Rule to issues positive feedback if the requirements are met.

It is also relevant to highlight that during the process of formalization, it happened that part of the recommendation was intrinsically difficult to model because of a vague formulation. For example, the above recommendation refers to the involvement of all major muscle groups, which is quite difficult to assess even when a domain expert is involved. As a result, for the time being, this aspect is not considered in our model.

## 3. Results

### 3.1. Description of HeLiFit Ontology

The development of the *HeLiFit* ontology has been a very ambitious task. The result is an ontology comprising both a taxonomically structured set of concepts and relations, as well as a set of rules defining complex relations, which define recommendations and the conditions under which they should be enabled.

The *HeLiFit* ontology is implemented in OWL 2 (Ontology Web Language) [52], using Protégé Desktop (version 5.5.0) and, it is provided as supplementary material.

It includes 250 classes, 13 Object Properties and 6 Datatype properties that formalize concepts and relationships to capture the underlying semantics of both the WHO Physical Activity & Sedentary Behaviour Guidelines [3] and the ACSM physical activity recommendations for the general adult population [53]. Overall, it includes 770 Logical Axioms (w.r.t. version V1.0). In relation to the kinds of axioms and class expressions used in HeLiFit, the underlying description logic is ALHI(D), whose complexity of concept satisfiability and ABox consistency lies between ALC (PSPACE) and SHOIQ (NExpTime), making the logic decidable [54]. In addition, the main ontology metrics are reported in Table 4, providing a complete picture of the extent of the entire model and its ontological entities. Finally, during the development process we made use of relevant Protege visualization plug-ins, such as VOWL (https://protegewiki.stanford.edu/wiki/VOWL, accessed on 28 November 2021) and OntoGraf(https://protegewiki.stanford.edu/wiki/OntoGraf, accessed on 28 November 2021), as well as OWLViz (https://protegewiki.stanford.edu/wiki/OWLViz, accessed on 28 November 2021) for deriving the ontology abstraction network, i.e., an algorithmically-derived summary of an ontology's structure and content, as presented and discussed in the next section. The current implementation (V1.0) is shared as Supplementary Materials.

**Table 4.** HeLiFit metrics.

| Metrics | | | |
|---|---|---|---|
| Axiom | 770 | Logical axiom count | 308 |
| Declaration axiom count | 289 | Class count | 250 |
| Object property count | 12 | Data property count | 5 |
| Individual count | 23 | Annotation property count | 1 |
| DL expressivity | ALHI(D) | SubClassOf | 246 |
| SubObjectPropertyOf | 2 | InverseObjectPropertiesf | 3 |
| ObjectPropertyDomain | 10 | ObjectPropertyRange | 10 |
| SubDataPropertyOf | 2 | DataPropertyDomain | 6 |
| DataPropertyRange | 6 | ClassAssertion | 23 |
| AnnotationAssertion | 173 | | |

### 3.2. Evaluation of HeLiFit and HeLiFit-Rule

**Validation.** We checked the logical consistency of classes, object properties and datatype properties inferences [55]. To achieve this, we applied directly the HermiT reasoner [55] (version 1.4.3) that is made available through Protege. The results proved that *HeLiFit* if free of logical inconsistencies or other errors that can be detected by the reasoner.

**Evaluation.** After validating the ontology for consistency, we proceeded with the evaluation of its appropriateness and usefulness with respect to the use cases of *Frailty and Sedentary Behaviour* and *Mental Health and Well-being*, in application scenario concerned with *Digital Coaching in the Healthcare Domain*. As a first step, the *HeLiFit* ontology was reviewed, during several sessions, by domain experts from the GATEKEEPER consortium, including Sport Medical Doctors and Mental Health Specialists, over the course of several sessions. During these sessions, the design rationale of the ontology, concepts and properties were explained, along with how to use *HeLiFit* to express the set of rules formalizing the recommendations. The domain experts found that the model is indeed consistent with a doctor's approach, when this provides advice to patients aiming to boost physical activity and reduce sedentary behavior. To evaluate the *usefulness* of *HeLiFit-Rule*, the domain expert proposed several standard patient profiles, to be assessed using *HeLiFit-Rule* as a decision support tool. Specifically, we constructed four exemplar user profiles, named *Sedentary* (Figure 12), *Moderate* (Figure 13), *Vigorous* (Figure 14) and *High* (Figure 15). Below we show the results obtained by applying our set of rules to these profiles.

In particular, for the *Sedentary* profile, we show in Figure 12 that Elisabeth, 54 years old, over a time window of a week, performed 1.5 h of walking (aerobic activities).

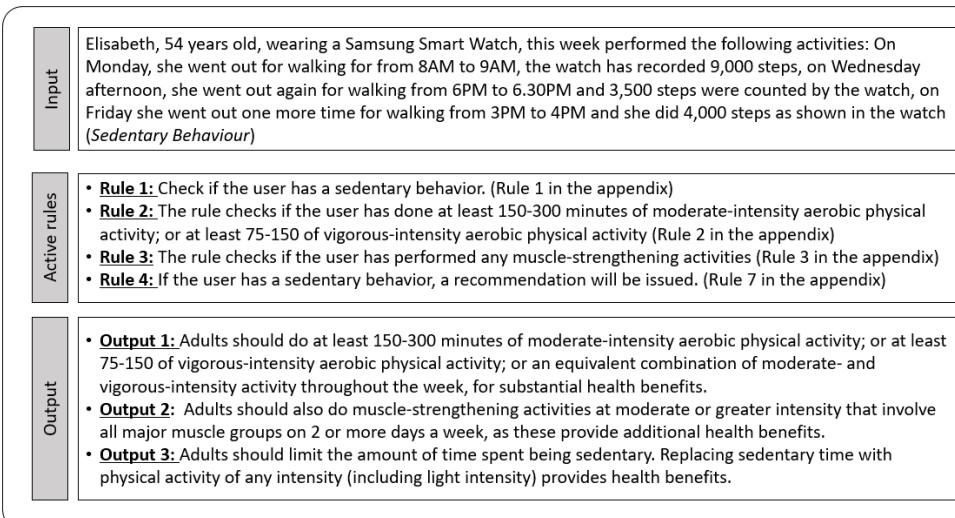

**Figure 12.** Sedentary profile.

Although she is compliant with the *duration* parameter, as 1.5 h is greater than the required 70 min, when it comes to *intensity*, this activity corresponds to a total of 4700 steps, which implies a very low speed. In this case, rule 5 (see Appendix A) is activated to detect this behavior and to classify Elisabeth as *Sedentary*. In addition, Elizabeth did not perform any muscle strengthening activity, hence, rule 17 (see Appendix A) is also triggered. As a result, the following recommendations are issued as reported in the output's section of Figure 12.

For *moderate* behavior, as illustrated in Figure 13, we consider Chris, who is 40 years old and has performed, over a time window of one week, 80 min of aerobic physical activity.

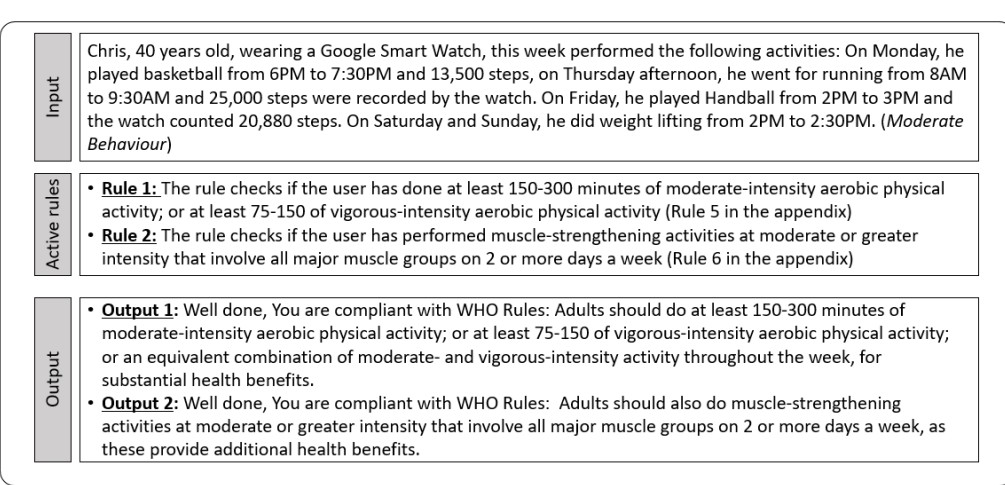

**Figure 13.** Moderate profile.

This level of activity is compliant with the WHO guidelines and, as a result, rule 13 is activated to issue a congratulatory message to Chris. However, he has performed muscle-strengthening only once, instead of the required two sessions a week, and therefore rule 15 is activated, which issues the following recommendations as reported in the output's section of Figure 13. For the *Vigorous* profile, we consider Juan (Figure 14), who is 25 years old. He has performed 105 min of vigorous aerobic physical activity, which is compliant with the WHO requirements. For this reason, a congratulatory message is issued by rule 12.

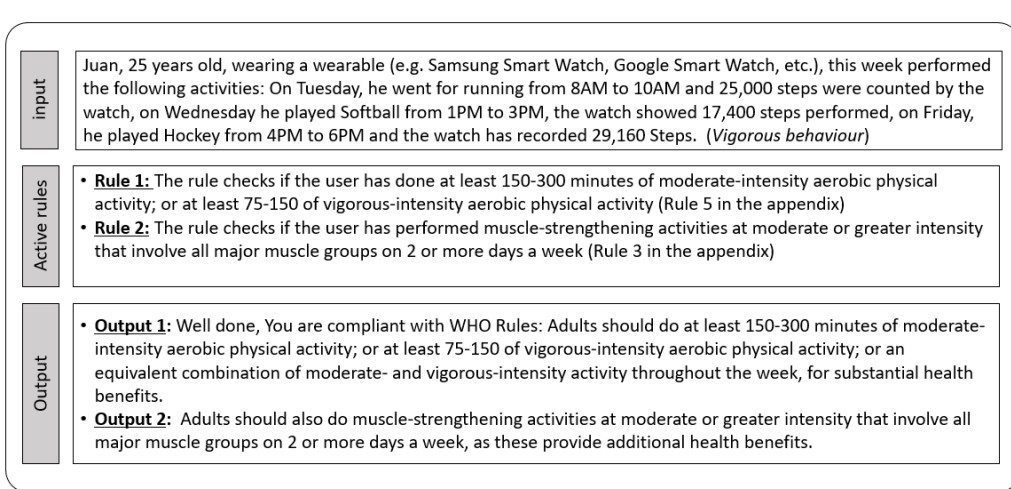

**Figure 14.** Vigorous profile.

However, Juan did not perform any muscle-strengthening activity, and therefore rule 17 is also activated, issuing the following recommendations as reported in the output's section of Figure 14. For the *high* profile, we assess Sara (Figure 15), who is 35 years old. Sara has performed 90 min of high aerobic activity and two muscle strengthening activities

during the week. Hence, she is compliant with the WHO recommendations and, as a result, rules 12 and 13 are activated to issue a congratulatory message as reported in the output's section of Figure 15.

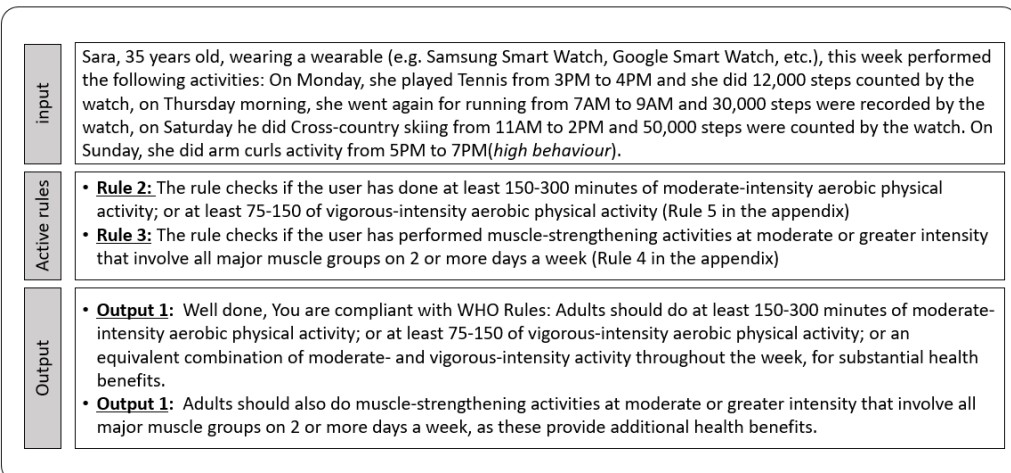

**Figure 15.** High profile.

To summarize, the domain experts stated that the application of the rules is consistent with their interpretation of the WHO/ACSM guidelines. They also added that, in their view, they can be effectively used in the context of an intervention plan that aims to help users to improve their lifestyle. Finally, the domain experts also stated that *HeLiFit-Rules* represents a resource of significant value, which can provide the basis for an appropriate support tool for both self-evaluation and external patient monitoring by professionals.

*3.3. Health Knowledge Graph with HeLiFit and Reasoning with Rules*

In this section we describe how the *HeLiFit* ontology is used operationally to build a Health Knowledge Graph, which is a necessary component of an application based on *HeLiFit-Rules*. Let's consider the following scenario:

Elisabeth, 54 years old, wearing a wearable (e.g., Smart Watch), this week performed the following activities: Monday, she went out for running for from 8 a.m to 9 a.m. and she did 10,000 steps Wednesday, she went out for a cycling session from 4 p.m to 5.30 p.m and she did 20 km, Friday, she went out for a swimming session from 7 p.m. to 7.45 p.m. and she did a total of 200 m. Based on this, the Digital Coach that she is using trough her mobile has to take a decision evaluating her overall performed activities, appropriately reward her and issue next suggestions according to guidelines.

To ensure a high level of syntactic and semantic interoperability across different systems, our framework is also compliant, especially on the input side, with the FHIR standard that were developed within GATEKEEPER project (https://build.fhir.org/ig/gatekeeper-project/gk-fhir-ig/, accessed on 28 November 2021).

We can use the *HeLiFit* ontology to represent these facts using the FHIR standard (Fast Healthcare Interoperability Resources is a standard describing data formats and elements and an application programming interface for exchanging electronic health records http://www.hl7.org/fhir/overview.html, accessed on 28 November 2021) and we can then use our set of rules to check adherence with the WHO/ACSM guidelines and issue recommendations tailored to individual circumstances. The details of this example are shown in Figure 16 in the form of a diagram using *HeLiFit* ontology and in Figure 17 as a Knowledge Graph using Turtle triple format.

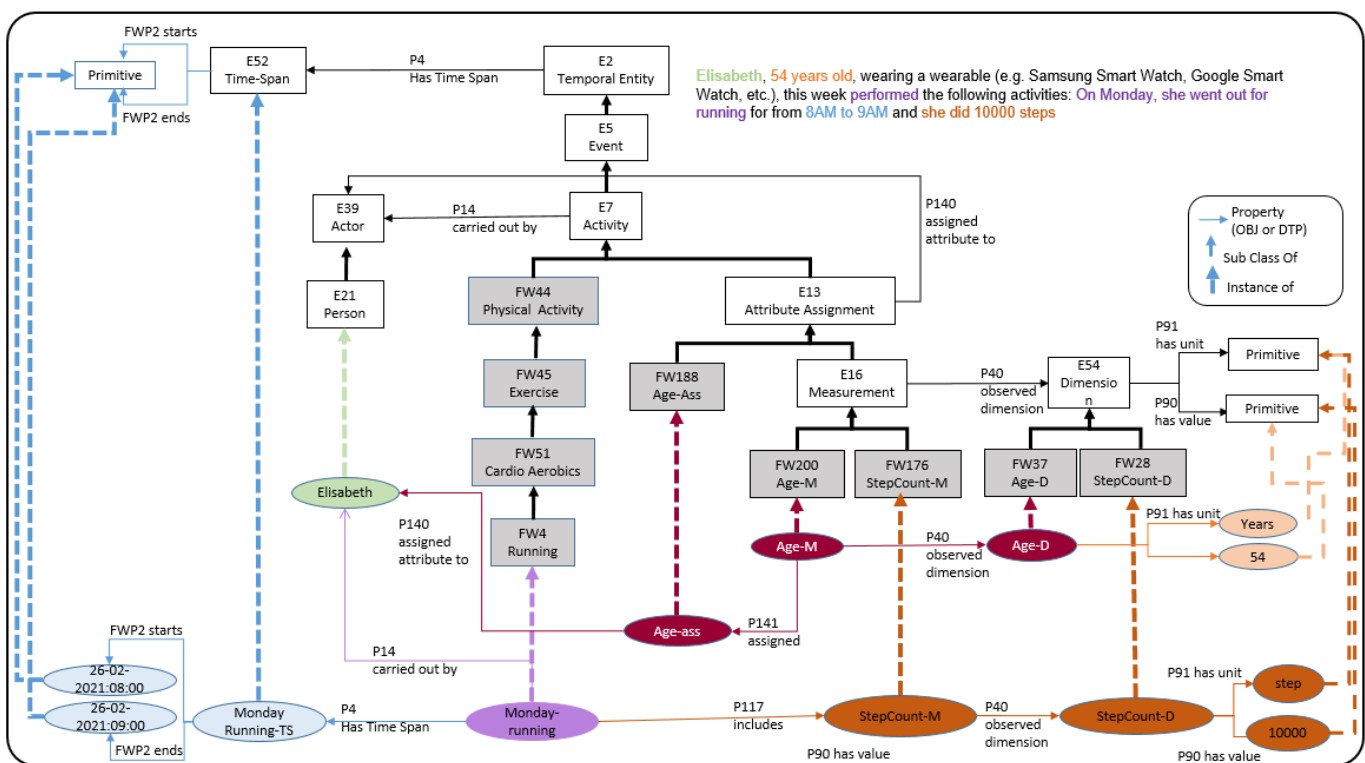

**Figure 16.** Instantiate HeLiFit over user's data.

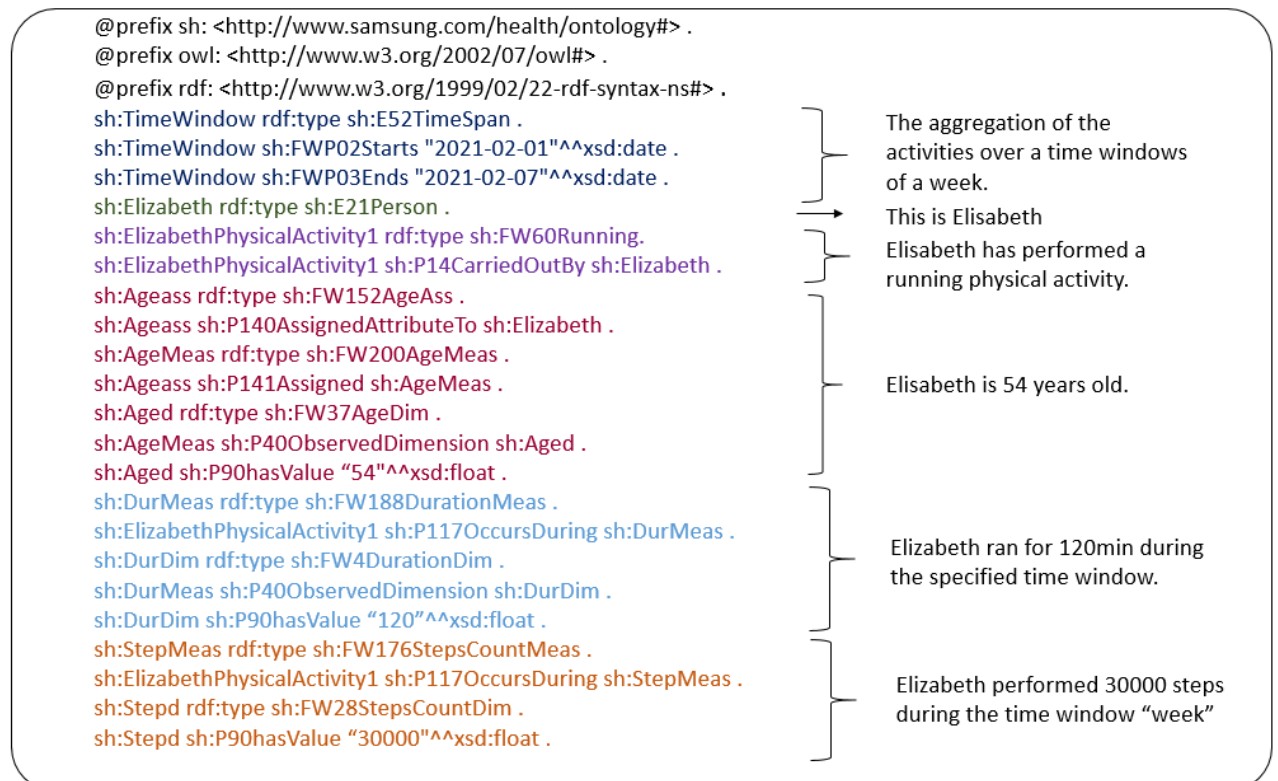

**Figure 17.** Knowledge Graph with HeLiFit over user's data.

As shown in Figure 18, each rule outputs a recommendation code that is formatted as an ontology URI; the one reported as an example above is *sh:WHOCode001*, with a prefix *sh* that corresponds to the one associated with the *HeLiFit* ontology. By doing so, we have structured the description of the WHO and ACSM recommendations as part of a larger

knowledge graph and, to this purpose, we used the schema shown in Figure 19 and its corresponding RDF codification, which is shown in Figure 20.

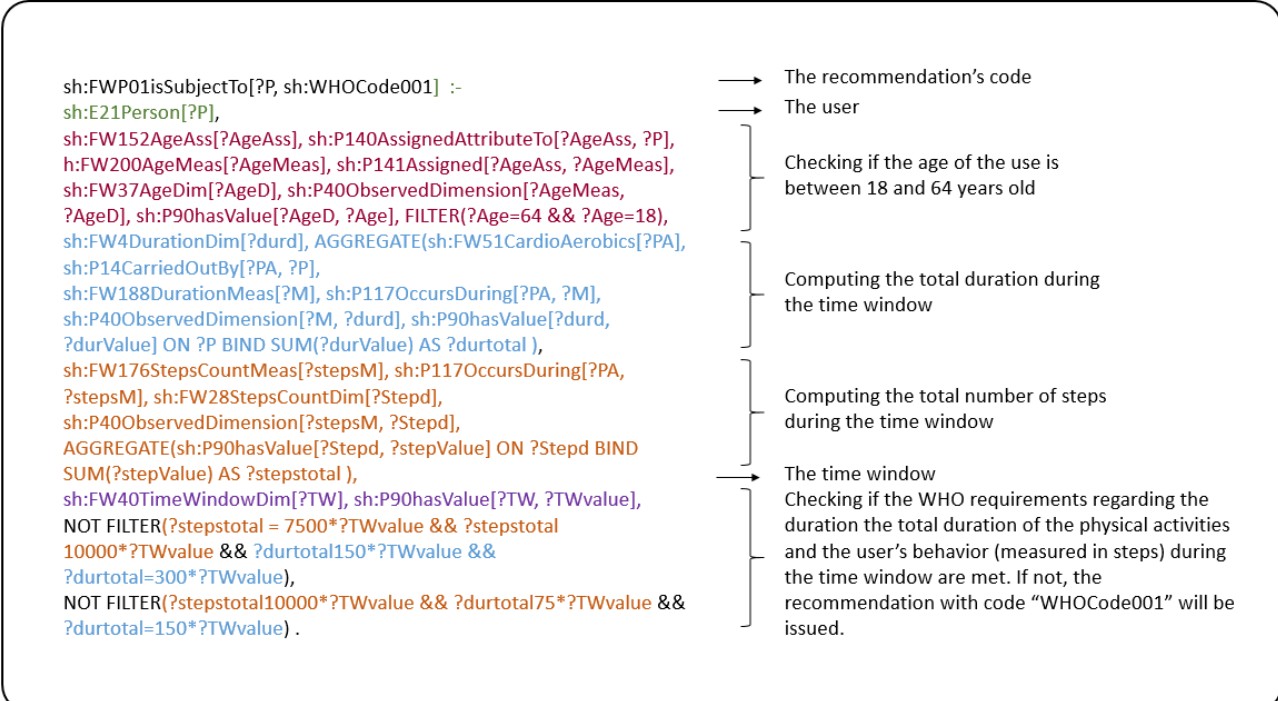

**Figure 18.** A rules codified using RDFox Rule Language.

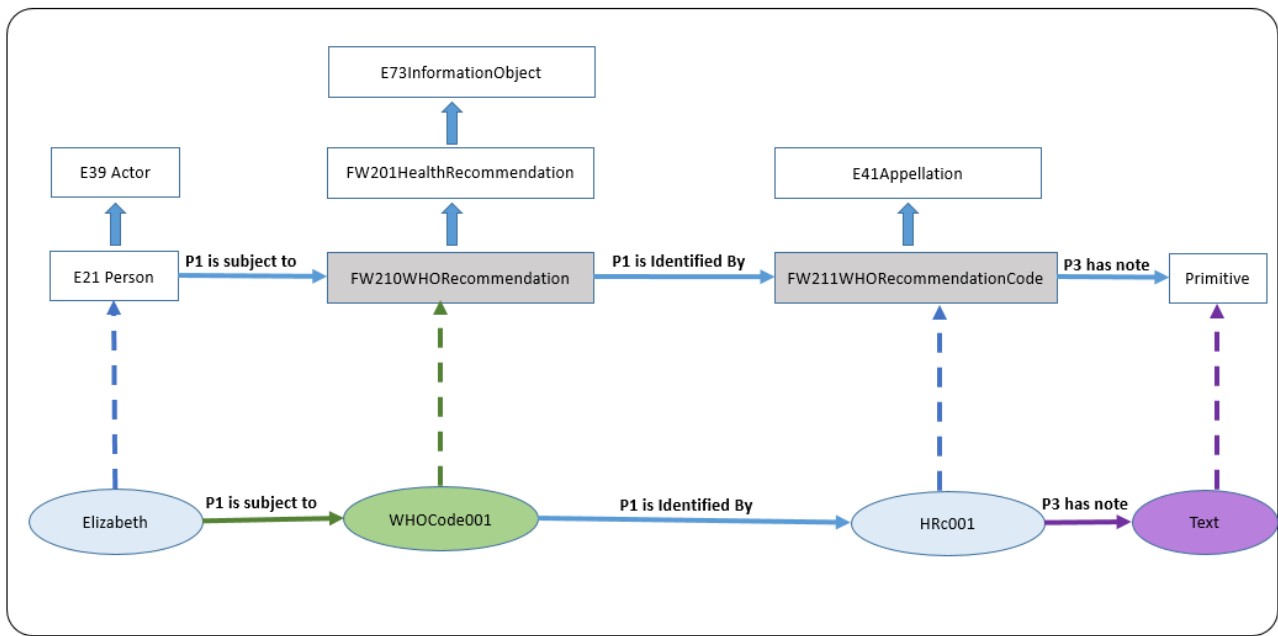

**Figure 19.** Recommendations' pattern.

```
@prefix sh: <http://www.samsung.com/health/ontology#> .
@prefix owl: http://www.w3.org/2002/07/owl: .
@prefix rdf: <http://www.w3.org/1999/02/22-rdf-syntax-ns#> .
sh:WHOCode001 rdf:type sh:FWE205WHOPhysicalActivityRecommendation .
sh:WHOCode001 sh:P1isIdentifiedBy sh:hrc1 .
sh:hrc1 rdf:type sh:FWE202WHORecommendationCode .
sh:hrc1 sh:P3hasNote "Adults should do at least 150-300 minutes of moderate-
intensity aerobic physical activity; or at least 75-150 of vigorous-intensity
aerobic physical activity; or an equivalent combination of moderate- and
vigorous-intensity activity throughout the week, for substantial health
benefits."^^xsd:string .
```

Recommendation code
Recommendation identifier
Recommendation description

**Figure 20.** Knowledge Graph with HeLiFit for representing WHO/ACSM recommendation.

This has the advantage that the rules provide a recommendation code, manually generated (e.g., WHOCode and ACSMCode), in order to distinguish the provenance between WHO and ACSM guidelines. Once the rules have computed the code, the corresponding text description is extracted from the knowledge graph. In order to retrieve it, we execute a specific SPARQL query with the pattern used to codify it. The code is shown in Figure 21:

```
SELECT distinct ?WHOrec ?textWHO ?p
WHERE {
?WHOrec rdf:type sh:FW205WHOPhysicalActivityRecommendation .
?WHOrec sh:P1isIdentifiedBy ?WHOrc .
?WHOrc rdf:type sh:FW202WHORecommendationCode .
?WHOrc sh:P3hasNote ?textWHO .
?p rdf:type sh:E21Person .
?p sh:FWP01isSubjectTo ?WHOrec .}
```

**Figure 21.** SPARQL code to issue the recommendation description.

### 4. Concluding Discussion and Future Work

**Outline of achieved objectives.** We developed the *HeLiFit* ontology and *HeLiFit-Rules* with the aim of formalizing the (WHO/ACSM) Physical Activity & Sedentary Behaviour Guidelines and apply these to specific user data. In particular, given a set of user data profiles, we can use *HeLiFit* to build a Knowledge Graph and *HeLiFit-Rules* to assess adherence to the WHO/ACSM guidelines and provide appropriate recommendations. *HeLiFit* and *HeLiFit-Rules*, as an integrated component, can serve as a knowledge tool for a Digital Coach, which aism to deal with unavoidable aspects of the ageing process: Frailty/Sedentary Behaviour and issues related to Mental Health and Wellbeing. From the perspective of knowledge engineering, *HeLiFit* and *HeLiFit-Rules* may be reused as a part of a broader solution, such as Digital Coach, to monitor and advise patients, by taking into account not only physical activities but also other dimensions including nutrition and food recommendations. Hence, it can be used to target patients with a variety of diseases, including Cancer Survivors, Depression, PTSD and Hypertension, just to mention a few, by tailoring a combination of exercise and food strategies. In this paper, we did not aim to develop a complete solution, for example a comprehensive Digital Coach, but we focused on ensuring that *HeLiFit* and *HeLiFit-Rules* can effectively provide adherence to guidelines and issue appropriate recommendations. In particular, the evaluation of *HeLiFit* and *HeLiFit-Rules* provides initial evidence that the formalization is able to assess user adherence to the WHO/ACSM guidelines, by considering her/his performances of

physical activities over a period of time (usually a week). In addition, both domain experts and users can use it to monitor the health of a user and promote lifestyle change.

**Interpretation of the results and future work.** The heterogeneous guidelines from ACSM and WHO were systematized through an approach based on the AI paradigm of ontology + rules. We focused on expressing the relations between four key concepts related to physical activity and/or exercise classification: (i) duration, (ii) frequency, (iii) modality and (iv) intensity. To describe these concepts and the corresponding relationships, we reused a top level ontology, CIDOC-CRM. In particular, we used the event-based *Activity-Temporal-Entity-Pattern* to capture the semantics of the physical activity as an event that happens over a limited extent in time and to model the other parameters associated with it, including those mentioned above. Furthermore, we described the relations among these concepts, by reusing the set of top level relationships provided by CIDOC-CRM. Given this ontological basis, we then formalized 153 rules that allow us to categorize user profiles according to their level of physical activity and issue appropriate recommendations in order to promote healthy behaviors. Needless to say, we are aware that *HeLiFit* does not necessarily cover every single aspect associated with the notion of physical activity. Our primary aim here was to cover the elements needed to model the WHO/ACSM guidelines. Having said so, we believe that it is generic enough to be extended to any level of detail on demand. Hence it could also be used to formalize other sets of recommendations for physical activity, including those specified by the American Diabetes Association (ADA), the Heart Failure Association (HFA) and the European Association for Cardiovascular Prevention and Rehabilitation (EACPR). These approaches provide specific recommendations for patients with diabetes or heart failure and fully endorse the use of a personalised, patient-centred approach to promote physical activity. Additionally co-morbidities regarding the mental state (e.g., eating disorders, obsessive compulsive disorders, major depression, PTSD) need to be taken into account when giving recommendations to avoid side effects and contraindications. Furthermore, the interoperability of *HeLiFit* can be improved by incorporating the compendium that was developed to facilitate the coding of physical activities on the basis of the rate of energy expenditure [56]. The purpose of such an extension is to be able to deal with and provide recommendations that consider a much larger spectrum of physical activities, including Home Activities, Lawn and Garden, Transportation, Water Activities and so on.

We recognize that our work present several limitations. Firstly, we have expressed our rules and their evaluation using the usual "true or false" Boolean logic. In our model, one could be considered to be sedentary even if he/she reached 69 min and 59 s of moderate-intensity aerobic physical activity, instead of 70 min, as per guidelines. Hence, we plan to extend our model by introducing a "fuzzy" interpretation that is based on "degrees of truth" rather than boolean logic. Specifically, we plan to investigate the feasibility of using a Fuzzy Logic approach to rule formalization, making use of formalisms such as SWRL-F [57]. By doing so, we believe that it will increase the practical value of our work, enabling domain experts to evaluate more precisely the impact of specific recommendations on user behavior. Secondly, to complement an approach to increasing physical activity and reducing sedentary behavior, a strong related perspective is the one of *Nutrition* [25], especially in the context of Digital Coach systems. In this respect, we intend to augment the ontology with information about Nutrition and Diet, including recipes to be recommended to users. The HeLiS ontology already shows the feasibility to link these two domains [25] and therefore will be considered as a starting point for this direction of research. We also aim to expand the ontology by considering the conceptual elements associated with mental health and well-being. Specifically, the expanded ontology will also take into account the current mental conditions of a person, such as depression, anxiety, social support and quality of life, and it will be also integrated with the International Classification of Functioning, Disability and Health (ICF) [5] (http://rssandbox.iescagilly.be/international-classification-of-functioning-disability-and-health.html, accessed on 28 November 2021) and of course adresses the security and privacy when deploying such a framework [58] .

**Supplementary Materials:** The following are available at https://www.mdpi.com/article/10.3390/app12041776/s1, Additional file S1: HeLiFit Ontology in OWL. Additional file S2: Domain Rule Set for formalizing the recommendation.

**Author Contributions:** Conceptualization, C.A.; methodology, C.A.; software, S.J., C.A., E.G. (Eleni Georga); validation, C.A., J.S. and L.S.; formal analysis, C.A.; investigation, C.A.; data curation, S.J. and C.A.; writing—original draft preparation, C.A., S.J.; writing—review and editing, C.A., S.J., E.M., A.A., M.S.H.and E.G. (Eleni Georga); visualization, C.A., S.J.; supervision, C.A.; project administration, R.A.; funding acquisition, R.A., G.F., L.P. and E.G. (Eugenio Gaeta). Proof-Reading, C.A., S.J., R.A., J.L., B.K., A.A., E.M., J.S., L.S., M.S.H., E.G. (Eleni Georga), L.P., E.G. (Eugenio Gaeta) and G.F. All authors have read and agreed to the published version of the manuscript.

**Funding:** This research has received funding from the European Union's Horizon 2020 research and innovation program under grant agreement No 857223.

**Institutional Review Board Statement:** Not applicable.

**Informed Consent Statement:** Not applicable.

**Data Availability Statement:** The *HeLiFit* ontology and *HeLiFit-Rule* are available as per extra documents attached to this file.

**Conflicts of Interest:** The authors declare that they have no competing interests.

## Abbreviations

The following abbreviations are used in this manuscript:

| | |
|---|---|
| WHO | World Health Organization |
| ACSM | The American College of Sports Medicine |
| OWL | Web Ontology Language |
| KR&R | Knowledge representation and reasoning |
| HeLiFit | Health Lifestyle Fitness Ontology |
| HeLiFit-Rule | Set of rules based on Health Lifestyle Fitness Ontology |
| SWRL | Semantic Web Rule Language |
| RDFox | A scalable in-memory RDF triple store and semantic reasoning engine |
| FHIR | Fast Healthcare Interoperability Resources |

## Appendix A. Set of Rules Used in This Paper

- *Rule 1:* *Checking if the user has a sedentary behavior based on the step count.*

```
sh:P2hasType[?P,sh:sedentarybehavior]:-sh:E21Person[?P],
sh:FW44PhysicalActivity[?PA],sh:P14CarriedOutBy[?PA,?P],
sh:FW176StepsCountMeas[?M],sh:P117Includes[?PA,?M],
sh:FW28StepsCountDim[?Stepd], sh:P40ObservedDimension[?M,?Stepd],
AGGREGATE(sh:P90hasValue[?Stepd,?stepValue]
ON ?Stepd BIND SUM(?stepValue) AS ?total) FILTER(?total<=5000) .
```

- *Rule 2:* *Checking if the user has done at least 150–300 min of moderate-intensity aerobic physical activity; or at least 75–150 of vigorous-intensity aerobic physical activity, if not, a recommendation will be issued.*

```
sh:FWP01isSubjectTo[?P,sh:WHOCode001]:-sh:E21Person[?P],
sh:FW152AgeAss[?AgeAss], h:P140AssignedAttributeTo[?AgeAss,?P],
sh:FW200AgeMeas[?AgeMeas],sh:P141Assigned[?AgeAss,?AgeMeas],
sh:FW37AgeDim[?AgeD],sh:P40ObservedDimension[?AgeMeas,?AgeD],
sh:P90hasValue[?AgeD,?Age],FILTER(?Age<=64&&?Age>=18),
sh:FW4DurationDim[?durd],
AGGREGATE(sh:FW51CardioAerobics[?PA],
sh:P14CarriedOutBy[?PA,?P],sh:FW188DurationMeas[?M],
sh:P117Includes[?PA,?M],sh:P40ObservedDimension[?M,?durd],
sh:P90hasValue[?durd,?durValue] ON ?P BIND SUM(?durValue)
AS ?durtotal),sh:FW176StepsCountMeas[?stepsM],
sh:P117Includes[?PA,?stepsM],sh:FW28StepsCountDim[?Stepd],
```

```
sh:P40ObservedDimension[?stepsM,?Stepd],
AGGREGATE(sh:P90hasValue[?Stepd,?stepValue] ON ?Stepd BIND
SUM(?stepValue) AS ?stepstotal ),sh:FW40TimeWindowDim[?TW],
sh:P90hasValue[?TW,?TWvalue],
NOT FILTER(?stepstotal >=7500*?TWvalue && ?stepstotal <10000*?TWvalue &&
?durtotal>150*?TWvalue && ?durtotal<=300*?TWvalue),NOT
FILTER(?stepstotal>10000*?TWvalue&&?durtotal>75*?TWvalue
&&?durtotal<=150*?TWvalue).
```

- *Rule 3: Checking if the user has performed any muscle-strengthening activities, if not, a recommendation will be issued.*

```
sh:FWP01isSubjectTo[?P,sh:WHOCode002]:-sh:E21Person[?P],
sh:FW152AgeAss[?AgeAss],
sh:P140AssignedAttributeTo[?AgeAss,?P],
sh:FW200AgeMeas[?AgeMeas],sh:P141Assigned[?AgeAss,?AgeMeas],
sh:FW37AgeDim[?AgeD],sh:P40ObservedDimension[?AgeMeas,?AgeD],
sh:P90hasValue[?AgeD,?Age],FILTER(?Age<=64&&?Age>=18),
sh:FW44PhysicalActivity[?PA],sh:P14CarriedOutBy[?PA,?P],NOT
EXISTS ?PA IN sh:FW49MuscleStrengtheningAnaerobics[?PA].
```

- *Rule 4: Checking if the user has performed muscle-strengthening activities at moderate or greater intensity that involve all major muscle groups on 2 or more days a week, if not, a recommendation will be issued.*

```
sh:FWP01isSubjectTo[?P,sh:WHOCode002]:-sh:E21Person[?P],
sh:FW152AgeAss[?AgeAss],sh:P140AssignedAttributeTo[?AgeAss,
?P],sh:FW200AgeMeas[?AgeMeas],sh:P141Assigned[?AgeAss,
?AgeMeas],sh:FW37AgeDim[?AgeD],
sh:P40ObservedDimension[?AgeMeas,?AgeD],
sh:P90hasValue[?AgeD, ?Age],FILTER(?Age<=64&&?Age>=18),
sh:FW2FrequencyDim[?freqD],
AGGREGATE(sh:FW49MuscleStrengtheningAnaerobics[?PA],
sh:P14CarriedOutBy[?PA, ?P],sh:FW193FrequencyMeas[?M],
sh:P117Includes[?PA, ?M],sh:P40ObservedDimension[?M, ?freqD],
sh:P90hasValue[?freqD,?freqValue] ON ?P BIND SUM(?freqValue)
AS ?freqtotal ),sh:FW176StepsCountMeas[?stepsM],
sh:P117Includes[?PA,?stepsM],sh:FW28StepsCountDim[?Stepd],
sh:P40ObservedDimension[?stepsM,?Stepd],
AGGREGATE(sh:P90hasValue[?Stepd,?stepValue] ON ?Stepd BIND
SUM(?stepValue) AS ?stepstotal ),sh:FW40TimeWindowDim[?TW],
sh:P90hasValue[?TW,?TWvalue],NOT FILTER(?stepstotal
>=7500*?TWvalue&&?freqtotal>=2*?TWvalue) .
```

- *Rule 5: Checking if the user has done at least 150–300 min of moderate-intensity aerobic physical activity; or at least 75–150 of vigorous-intensity aerobic physical activity, if so, a congratulations message will be issued.*

```
sh:FWP01isSubjectTo[?P,sh:WHOCode101]:-sh:E21Person[?P],
sh:FW152AgeAss[?AgeAss],sh:P140AssignedAttributeTo[?AgeAss,?P],
sh:FW200AgeMeas[?AgeMeas],sh:P141Assigned[?AgeAss,?AgeMeas],
sh:FW37AgeDim[?AgeD],sh:P40ObservedDimension[?AgeMeas,?AgeD],
sh:P90hasValue[?AgeD,?Age],FILTER(?Age<=64&&?Age>=18),
sh:FW4DurationDim[?durd],AGGREGATE(sh:FW51CardioAerobics[?PA],
sh:P14CarriedOutBy[?PA,?P],sh:FW188DurationMeas[?M],
sh:P117Includes[?PA,?M],sh:P40ObservedDimension[?M,?durd],
sh:P90hasValue[?durd,?durValue] ON ?P BIND SUM(?durValue)
AS ?durtotal),sh:FW176StepsCountMeas[?stepsM],
sh:P117Includes[?PA,?stepsM],sh:FW28StepsCountDim[?Stepd],
sh:P40ObservedDimension[?stepsM,?Stepd],
AGGREGATE(sh:P90hasValue[?Stepd,?stepValue] ON ?Stepd
BIND SUM(?stepValue) AS ?stepstotal),
sh:FW40TimeWindowDim[?TW],sh:P90hasValue[?TW,?TWvalue],
```

```
FILTER(?stepstotal>=7500*?TWvalue && ?stepstotal<10000*?TWvalue
&&?durtotal>150*?TWvalue&&?durtotal<=300*?TWvalue).
```

- ***Rule 6:*** *Checking if the user has performed muscle-strengthening activities at moderate or greater intensity that involve all major muscle groups on 2 or more days a week, if so, a congratulations message will be issued.*

```
sh:FWP01isSubjectTo[?P,sh:WHOCode102]:-sh:E21Person[?P],
sh:FW152AgeAss[?AgeAss],sh:P140AssignedAttributeTo[?AgeAss,?P]
sh:FW200AgeMeas[?AgeMeas],sh:P141Assigned[?AgeAss,?AgeMeas],
sh:FW37AgeDim[?AgeD],sh:P40ObservedDimension[?AgeMeas,?AgeD],
sh:P90hasValue[?AgeD,?Age],FILTER(?Age<=64&&?Age>=18),
sh:FW2FrequencyDim[?freqD],
AGGREGATE(sh:FW49MuscleStrengtheningAnaerobics[?PA],
sh:P14CarriedOutBy[?PA,?P],sh:FW193FrequencyMeas[?M],
sh:P117Includes[?PA,?M],sh:P40ObservedDimension[?M,?freqD],
sh:P90hasValue[?freqD,?freqValue] ON ?P BIND
SUM(?freqValue) AS?freqtotal),sh:FW176StepsCountMeas[?stepsM],
sh:P117Includes[?PA,?stepsM],sh:FW28StepsCountDim[?Stepd],
sh:P40ObservedDimension[?stepsM,?Stepd],
AGGREGATE(sh:P90hasValue[?Stepd,?stepValue] ON ?Stepd BIND
SUM(?stepValue) AS ?stepstotal),sh:FW40TimeWindowDim[?TW],
sh:P90hasValue[?TW,?TWvalue],FILTER(?stepstotal>=7500*?TWvalue
&& ?freqtotal>=2*?TWvalue) .
```

- ***Rule 7:*** *Checking if the user has a sedentary behavior, if so, a recommendation will be issued.*

```
sh:FWP01isSubjectTo[?P,sh:WHOCode005]:-sh:E21Person[?P],
sh:FW152AgeAss[?AgeAss],
sh:P140AssignedAttributeTo[?AgeAss,?P],
sh:FW200AgeMeas[?AgeMeas],sh:P141Assigned[?AgeAss,?AgeMeas],
sh:FW37AgeDim[?AgeD],sh:P40ObservedDimension[?AgeMeas,?AgeD],
sh:P90hasValue[?AgeD,?Age],FILTER(?Age<=64&&?Age>=18),
sh:P2hasType[?P,sh:sedentarybehavior].
```

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
