# Peer review of "Toward a Symbolic AI Approach to the WHO/ACSM Physical Activity & Sedentary Behavior Guidelines"

_applsci, doi:10.3390/app12041776_

Round 1
Reviewer 1 Report
In this work, authors design and implement an ad-hoc ontology aiming to formalize the Physical Activity and Sedentary Behaviour Guidelines, confronting applying these to specific user data. Given a set of user data profiles, they are able to build a Knowledge Graph and Rules to assess adherence to the WHO/ACSM guidelines and provide appropriate recommendations.
The proposed approach is really interesting.
The described methodology is sound and well-illustrated. The work is easily understandable, English is good. Finally, the challenges and future solutions are pointed out for power inspection.
I propose that this paper will be accepted for final publication.
Reviewer 2 Report
The paper describes a formalization approach for physical activity and sedentary behaviour guidelines based on ontologies and rules. The overall goals are well motivated and the paper is well written and understandable. I am not a healthcare expert , but could easily follow the paper and its argumentation. The overall approach seems to be straight forward referring to the common standards and tools of the sematic web domain. In my opinion , the paper is a bit lengthy and some of the examples and described user profiles could be deleted without a significant loss of information to the reader. I have some suggestions for improving the paper but no substantial critics. Therefore , I recommend a minor revision of the paper.
Suggestions to improve the paper:
- „Samsung’s Digital Coach“ has been mentioned several times as an example application in the paper. This relates to the affiliation of some of the authors and feels a little like an advertisement. Please , also provide other example systems and delete the term „Samsung’s Digital Coach“ in the cases where it is not really required. In my opinion, it should not be mentioned more than 2 times in the paper. It is also insignificant if one of the profiles wears a Samsung smart watch. This could also be a device any other vendor.
- Subchapters 2.1.2 and 2.1.4 describe the same goals and could be integrated , in my opinion.
- For understandability , figures 5 and 6 should contain all classes used in the following rules.
- The rules and their formalization are difficult to read and understand. For readers not aware of the used rule formalization a short explanation should be added to the description of Rule A. Further, some commenting in the rule code would be helpful (as it has been done in figure 12), as the terms in the formalized rules do not always directly contain the terms of the verbal description.
- The outputs described in Lines 506 till 516 are also included in figure 7 and can be deleted from the text. The same applies to the output descriptions for the following figures. Are all 4 profiles really necessary? In my opinion, one profile would be sufficient, as the others do not bring any substantial additional information to the reader.
- Is figure 11 necessary for understanding? This is all contained in the text before. Figure 12 is not referenced from the text.
Some minor things:
- Line 71 : “..”
- Table 1: the caption does not fit here
- Some of the figures are difficult to read; especially the text in figure 11 and 14 must be enlarged.
- Line 484: “The goal is….” Something wrong with this sentence.
Reviewer 3 Report
The Paper is well written and contributes towards the very pressing issue of homecare and elderly care. I hope the project result will be freely available to the research community through Bioportal. Paper need to be checked throughout to remove double full stop here and there. Figure 12. "Knowledge Graph with HeLiFit over user’s data." is a code but I was expecting it on Graph visualization.
It will be great if the Author showcase how their model is compatible with other Smart watch application or how much we can reuse. Incase of ontology it has many shortcomings e.g. Annotation is not defined for almost all classes or properties.
